# Protective role of fructokinase blockade in the pathogenesis of acute kidney injury in mice

Ana Andres-Hernando[1,*], Nanxing Li[1,*], Christina Cicerchi[1,*], Shinichiro Inaba[1], Wei Chen[1,2], Carlos Roncal-Jimenez[1], Myphuong T. Le[1], Michael F. Wempe[1], Tamara Milagres[1], Takuji Ishimoto[1], Mehdi Fini[1], Takahiko Nakagawa[1], Richard J. Johnson[1] & Miguel A. Lanaspa[1]

Acute kidney injury is associated with high mortality, especially in intensive care unit patients. The polyol pathway is a metabolic route able to convert glucose into fructose. Here we show the detrimental role of endogenous fructose production by the polyol pathway and its metabolism through fructokinase in the pathogenesis of ischaemic acute kidney injury (iAKI). Consistent with elevated urinary fructose in AKI patients, mice undergoing iAKI show significant polyol pathway activation in the kidney cortex characterized by high levels of aldose reductase, sorbitol and endogenous fructose. Wild type but not fructokinase knockout animals demonstrate severe kidney injury associated with ATP depletion, elevated uric acid, oxidative stress and inflammation. Interestingly, both the renal injury and dysfunction in wild-type mice undergoing iAKI is significantly ameliorated when exposed to luteolin, a recently discovered fructokinase inhibitor. This study demonstrates a role for fructokinase and endogenous fructose as mediators of acute renal disease.

[1] University of Colorado Denver, 12700 East 12th Ave C-281, Aurora, Colorado 80045, USA. [2] Department of Nephrology, The First Affiliated Hospital, Sun Yat-sen University, Guangzhou, Guangdong 510080, China. * These authors contributed equally to this work. Correspondence and requests for materials should be addressed to M.A.L. (email: miguel.lanaspagarcia@ucdenver.edu).

Acute kidney injury (AKI) is a common clinical syndrome that complicates up to −20% of hospital admissions and 30–50% of intensive care unit admissions[1,2]. AKI is associated with up to an eight-fold increase risk in mortality[3,4] and ischaemia is one of the most common causes of AKI accounting for 50% of all cases. Patients who develop AKI stay in the hospital longer[2] and are twice as likely to be discharged to short- or long-term care facilities[5]. Of interest, after decades of important discoveries regarding its pathophysiology, no clinically available treatment to accelerate kidney recovery in AKI has emerged and management is limited to supportive care, such as renal replacement therapy.

The polyol pathway is a molecular route constituted by two enzymes, aldose reductase and sorbitol dehydrogenase. The purpose of this pathway is the generation of sorbitol and fructose from glucose. In most tissues including the kidney cortex, this pathway is not active because aldose reductase is not expressed. However, when aldose reductase expression is upregulated, sorbitol and fructose (endogenous fructose) is significantly produced and metabolized[6,7]. To date, hypertonicity[8,9], hyperglycaemia[10] and hypoxia (ischaemia)[11,12] are the most important factors that stimulate aldose reductase expression in multiple tissues. We have previously shown that the activation of this pathway is an important deleterious step in the pathogenesis of multiple chronic diseases, including fatty liver[7] and chronic kidney disease[6]. However, to date the characterization of the potential deleterious role of endogenous fructose production and metabolism in AKI remains unknown.

The first step in fructose metabolism is mediated by fructokinase. Fructokinase phosphorylates fructose to fructose-1-phosphate. In most tissues, this step results in further metabolism of fructose-1-phosphate producing toxic advanced glycation end-products[13,14], induction of *de novo* fat synthesis and accumulation[15,16] and the induction of a marked ATP depletion[17,18]. Depleted ATP results in AMP accumulation and the rise in intracellular uric acid. Our published data[6,19,20] demonstrate that uric acid, while it is a well-known antioxidant in the extracellular environment, acts as a potent prooxidant molecule inside the cell triggering the generation of oxidative stress and causing cell death. Furthermore, blockade of fructokinase expression in renal proximal tubular cells inhibits fructose-induced production of oxidative stress and cell injury[20].

In this manuscript, we test the novel hypothesis that endogenous fructose production generated by the polyol pathway is a deleterious mechanism for causing ischaemic AKI (iAKI). Therefore, the blockade of this pathway could be clinically relevant not only as means to prevent iAKI (such as in cardiovascular surgery) but also as a target to accelerate renal recovery after the onset of renal injury.

## Results

**Increased urinary fructose levels in AKI patients.** Activation of the polyol pathway in human patients with AKI might be reflected by a significant increase in urinary fructose levels associated with significant tubular injury. To test this, urinary levels of fructose—corrected to urinary creatinine levels—were analysed in paediatric patients undergoing cardiac bypass surgery (CBP)[21]. The AKI group was defined by a 50% increase in serum creatinine at 24 h post surgery. As shown in Fig. 1, urinary fructose levels were significantly elevated at 6 h post-CBP in patients with AKI compared with CBP-no AKI patients (148.5 ± 59.25 in CBP-no AKI versus 646.1 ± 439.2 nmol fructose per UCre in CBP-AKI, $P = 0.0323$, two-tail $t$-test), indicating activation of the polyol pathway and the production of endogenous fructose in AKI. Of interest, no significant fructokinase activity was detected in urine aliquots.

**Fructose production in kidney cortex of mice with iAKI.** iAKI was performed in wild-type mice by clamping both renal pedicles for 22 min as detailed in the Methods section. To determine the activation of the polyol pathway and the onset of aldose reductase expression in the kidney cortex post-iAKI, mice were killed at different time points (baseline, 1, 2, 4, 8 and 24 h) and kidneys were removed for analysis. As shown in Fig. 2a, and consistent with previous reports[6,22,23], little or no expression of aldose reductase was found in the kidney cortex of mice at baseline. In contrast, aldose reductase levels were found to be gradually upregulated post iAKI in a time-dependent manner as demonstrated by western blot. This upregulation peaked at 8 h and was sustained until 24 h post-iAKI. While fructokinase expression is already present in the kidney cortex at baseline, its levels were higher in mice post-AKI in a time-dependent manner.

To better determine the site of expression of aldose reductase in the kidney cortex after iAKI, we analysed its location in frozen kidney tissues by confocal immunofluorescence. As seen in Fig. 2b, and consistent with our western blot data, aldose reductase (green) expression is upregulated at 24 h post-iAKI in wild-type mice. As shown in the image, there is a heterogeneous pattern of cytosolic expression of aldose reductase with proximal tubules expressing high amounts (denoted in the image with blue arrows) while other tubules expressed reduced (green arrows) or

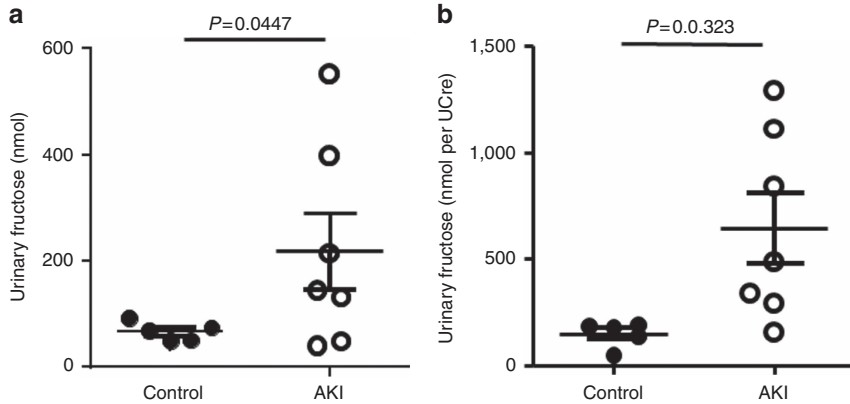

**Figure 1 | Elevated urinary fructose excretion in human subjects with acute kidney injury (AKI).** Urinary fructose levels total (**a**) or normalized to creatinine (**b**) in human subjects undergoing acute kidney injury ($n = 5–7$). Each sample was measured in triplicates and average for each subject was calculated for statistical analysis and displayed in the figure.

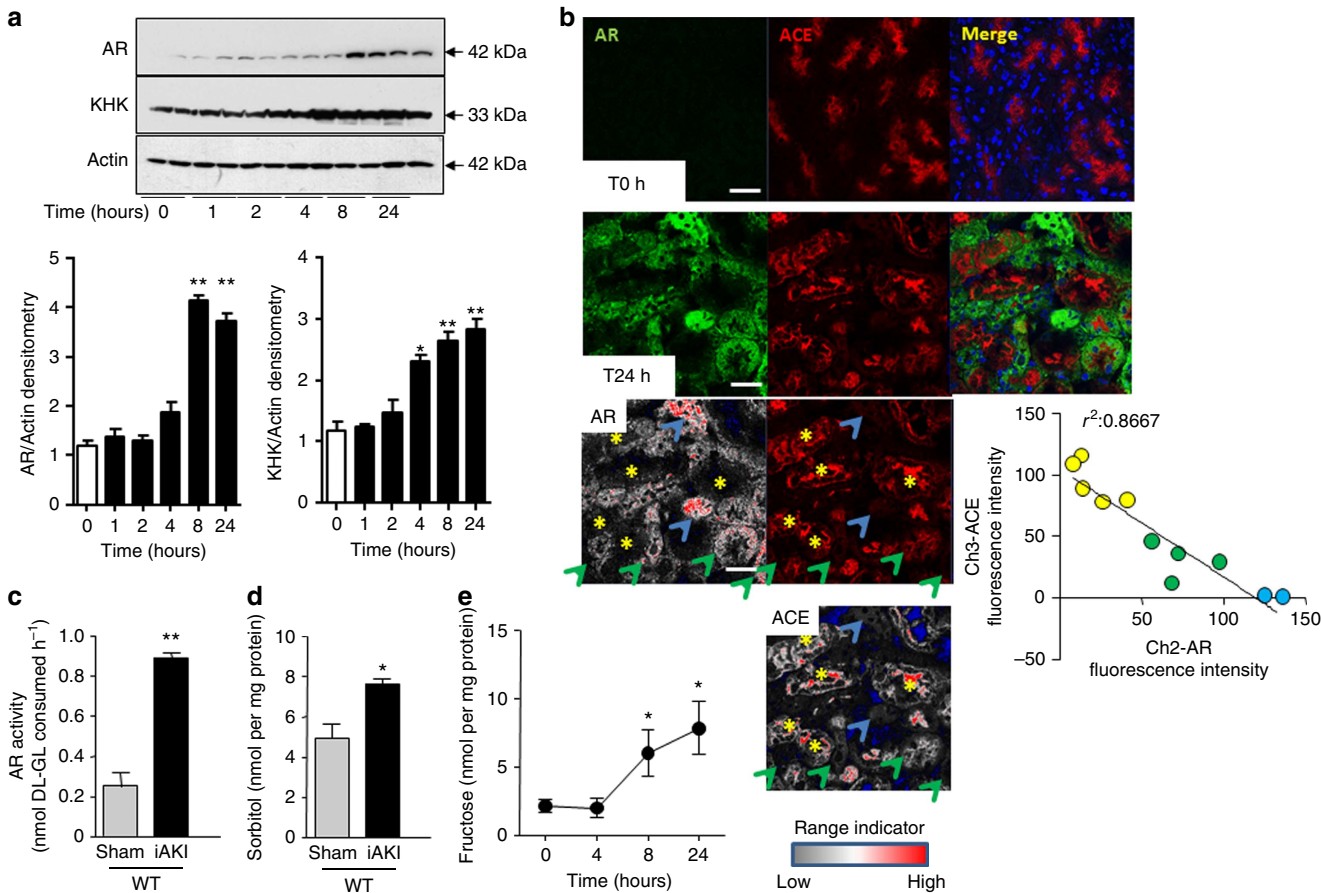

**Figure 2 | Activation of the polyol pathway in the kidney cortex of mice undergoing iAKI.** (**a**) Protein expression and densitometry of aldose reductase (AR) and fructokinase (ketohexokinase, KHK) in kidney cortex extracts at different time points post-ischaemic insult (uncropped blots is presented as Supplementary Fig. 1). Statistical analysis was performed with ANOVA with *ad hoc* analysis by Bonferroni's method to compare all columns. (**b**) Top: Representative fluorescent staining of the kidney cortex for aldose reductase (AR, green) and the brush border marker ACE (red) at baseline and 24 h post-ischaemic insult. Bottom: Confocal quantitation and plot for AR and ACE staining. Blue arrows indicate tubules with high AR/low ACE expression, while yellow asterisks represent tubules with low AR/high ACE expression. Scale bar: 100 μm. (**c**) Cortical AR activity at baseline and 24 h post-ischaemic insult (two-tailed *t*-test statistical analysis). (**d**) Cortical sorbitol levels at baseline and 24 h post-ischaemic insult (two-tailed *t*-test statistical analysis). (**e**) Time-course analysis of fructose levels in mice undergoing iAKI for 24 h (ANOVA with *ad hoc* analysis by Bonferroni's method to compare all columns). $n = 6$ animals per group with two different studies. Data indicate mean s.e.m. *$P < 0.05$, **$P < 0.01$ versus baseline or control.

no levels (yellow asterisks). Of interest, we observed an inverse correlation (Fig. 2b, bottom) between aldose reductase expression and brush border integrity as determined by angiotensin converting enzyme (ACE) staining (red), indicating that aldose reductase levels in the proximal tubule correlates with tubular damage.

Consistent with an elevation of aldose reductase expression in the kidney cortex, aldose reductase activity (Fig. 2c) as well as cortical levels of sorbitol (Fig. 2d) and fructose (Fig. 2e) was significantly elevated in mice undergoing iAKI in a time-dependent manner.

**Fructokinase deficiency exerts protection against iAKI.** To determine the specific role of the fructose that is being generated by the polyol pathway during iAKI, we blocked its metabolism using fructokinase-deficient mice. To this end, wild type and fructokinase-deficient mice underwent sham operation or iAKI and renal function and injury was assessed. Renal cortical levels of sorbitol and fructose at 24 h were not significantly different between wild type and fructokinase knockout mice suggestive of similar polyol pathway activation in both strains upon ischaemic insult (sorbitol: $7.2 \pm 0.3$ nmol mg$^{-1}$ in wild-type mice

versus $7.8 \pm 0.4$ nmol mg$^{-1}$ in mice deficient for fructokinase, and fructose: $7.3 \pm 2.6$ nmol mg$^{-1}$ in wild-type mice versus $8.2 \pm 3.4$ nmol mg$^{-1}$ in mice deficient for fructokinase). As depicted in Fig. 3a, left, serum creatinine levels are elevated in mice undergoing iAKI in a time-dependent manner. Of interest, this elevation in creatinine starts to be significant as early as 8 h post-insult in wild type but not in fructokinase-deficient mice, suggesting that the blockade of fructose metabolism slows the development of kidney injury. The elevation of creatinine levels at 8 h in wild type but not fructokinase-deficient mice undergoing iAKI is consistent with our previous observation (shown in Fig. 2a) in that the activation of the polyol pathway in the kidney cortex occurred at 8 h post-iAKI in mice. At 24 h post-iAKI serum creatinine levels are elevated in both strains compared with sham operation (Fig. 3a, right) although overall levels are significantly lower in fructokinase-deficient mice compared with wild-type animals. Blood urea nitrogen (BUN) levels, another marker of renal function, was also found to be significantly lower in fructokinase-deficient mice undergoing iAKI compared with wild type (Fig. 3b).

Wild-type mice developed worse renal injury than fructokinase-deficient mice as denoted by tubular dilatation and cast formation in periodic acid-Schiff -stained sections (Fig. 4).

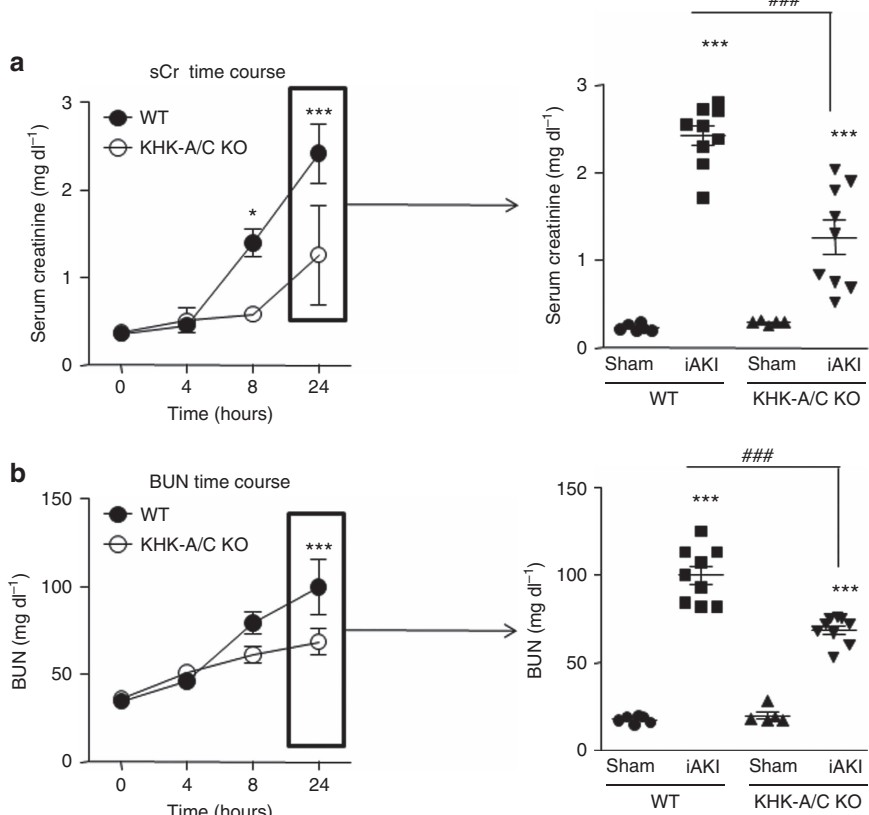

**Figure 3 | Ameliorated renal dysfunction in fructokinase knockout mice undergoing iAKI.** (**a**) Left: time-course analysis of serum creatinine (sCr) levels in wild type (WT) and fructokinase-deficient (KHK-A/C KO) mice undergoing iAKI for 24 h. Right: serum creatinine levels at 24 h in the same animals or sham control. (**b**) Left: time-course analysis of blood urea nitorgen (BUN) levels in wild type (WT) and fructokinase-deficient (KHK-A/C KO) mice undergoing iAKI for 24 h. Right: serum creatinine levels at 24 h in the same animals or sham control. $n = 5$–9 animals per group with two different studies. Statistical analysis was performed with ANOVA with *ad hoc* analysis by Bonferroni's method to compare all columns. Data indicate mean ± s.e.m. \*\*\*$P < 0.001$ versus baseline or within the same group. ###$P < 0.001$.

Furthermore, fructokinase-deficient mice had less brush border loss compared with wild-type animals as determined by the expression of ACE in the renal cortex[6] (Fig. 5a,b), and by the urinary levels of the tubular injury marker NGAL (Fig. 5c).

**Low inflammation and ATP loss in fructokinase null mice with iAKI.** Ischaemia is associated with reduced ATP levels and the build-up of uric acid and oxidative stress. Since the metabolism of fructose by fructokinase also induces ATP depletion[24] and uric acid generation[25], we hypothesized that fructokinase deficiency would result in increased ATP levels with reduction of local uric acid generation and oxidative stress in the kidneys of mice undergoing iAKI. As shown in Fig. 6a–c, ischaemia results in the rapid decrease of ATP levels within the first 4–8 h post insult in both wild type and fructokinase-deficient mice. Reduced ATP levels correlate with the building up of ADP, uric acid and the increase in ADP to ATP ratio. Consistent with an activation of fructokinase in wild-type mice, at 8 h post-iAKI renal ATP levels start to be restored back to baseline in fructokinase deficient but not in wild-type mice. This rapid restoration of renal ATP levels in fructokinase-deficient mice is accompanied by a reduction in ADP, uric acid and the ADP to ATP ratio in these mice which is not observed in wild-type animals until 24 h post-ischaemic insult. Reduced renal uric acid (Fig. 6d) levels correlated with decreased oxidative stress in proximal tubular cells as denoted by reduced dihydroethidium staining (Fig. 6e) and levels of thiobarbituric acid reactive substances (Fig. 6f).

We and others have previously shown that the metabolism of fructose and the generation of uric acid result in an inflammatory response which is mediated by the activation of the transcription factor NF-κB in pancreatic islets[26], liver[7] and kidney cells[20,27]. Therefore, we tested if the renal inflammation previously reported in mice with iAKI[28,29] could be mediated by fructose and uric acid. As shown in Fig. 7a, in wild-type mice undergoing iAKI there is a translocation of the catalytic subunit of NF-κB, p65, to the nucleus at 8 and 24 h post-insult, which is not observed in fructokinase-deficient mice. Consistent with higher nuclear expression levels of p65 in wild-type animals, the cytosolic levels of IκBα are much lower compared with fructokinase-deficient mice. Consistent with this observation, mRNA expression of the pro-inflammatory cytokine *il-6* and the chemokine *ccl2* were significantly higher in wild type compared with fructokinase-deficient mice. Conversely, fructokinase-deficient mice demonstrated greater renal expression of the anti-inflammatory cytokine *il-10* (Fig. 7b).

**Fructokinase inhibition protects mice from iAKI.** Luteolin is a flavone found in the leaves, barks and pollen of plants that has recently shown to exert protective effects against several forms of kidney disease, including diabetic nephropathy[30] and cisplatin-induced kidney injury[31]. Using an specific fructokinase activity assay based on ATP readout after fructose load (as in ref. 32), we have observed that luteolin is a potent fructokinase inhibitor *in vitro* (IC$_{50}$: 11.2 μM) and in human proximal tubular cells that express fructokinase (Fig. 8a). To test the effectiveness of

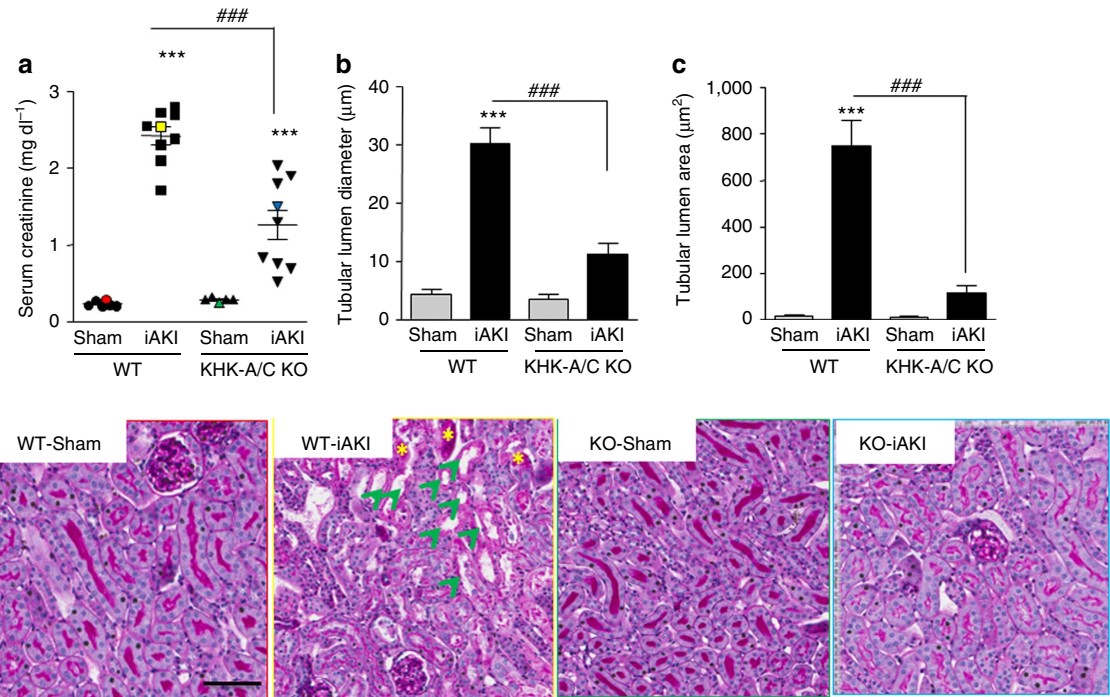

**Figure 4 | Ameliorated renal injury in fructokinase knockout mice undergoing iAKI.** (**a**) Serum creatinine and representative periodic acid-Schiff-stained images of sham operation and mice (wild type (WT) or fructokinase deficient (KHK-A/C KO) undergoing iAKI) (colour denotes animal selected from each group). (**b**) Tubular lumen diameter in the same groups of animals. (**c**) Tubular lumen area in the same groups of animals. Data indicate mean ± s.e.m. ***P < 0.001 versus baseline or within the same group. n = 5–9 animals per group with two different studies. ###P < 0.001. Green arrows denote tubules with significant dilatation. Statistical analysis was performed with ANOVA with *ad hoc* analysis by Bonferroni's method to compare all columns. Yellow asterisks: cast formation in tubes. Scale bar: 100 μm.

fructokinase inhibition, luteolin (2.5 mg kg$^{-1}$) was administered intravenously to wild-type mice at 90 min before, and 3 and 6 h post-ischaemic AKI insult——right before the polyol pathway is activated in the proximal tubule—and renal function and injury assessed. As shown in Fig. 8b, serum creatinine and BUN levels were significantly lower in luteolin-treated mice undergoing iAKI compared with vehicle (0.26 ± 0.11 mg dl$^{-1}$ in luteolin-treated group versus 1.42 ± 0.29 mg dl$^{-1}$ in vehicle-treated mice for creatinine; and 40.33 ± 16.04 mg dl$^{-1}$ in luteolin-treated group versus 103 ± 18.08 mg dl$^{-1}$ in vehicle-treated mice for BUN). Consistent with improved renal function, renal injury assessed by histology and urinary NGAL levels was dramatically reduced in luteolin-treated group (Fig. 8c–e).

When luteolin was administered after the ischaemic insult (30 min and 2 h post-ischaemia, Fig. 9), we observed that luteolin induced a significant elevation in urinary fructose levels when in a dose-dependent manner (Fig. 9a,b) demonstrating efficient inhibition of fructokinase. Consistent with a dose-dependent inhibition of fructokinase, a dose-dependent effect of luteolin in renal injury and dysfunction was determined (Fig. 9c), thus suggesting that the pharmacological inhibition of fructokinase may be clinically relevant not only to prevent kidney injury but also to accelerate kidney recovery post-iAKI.

**Fructokinase loss protects mice from CIN.** Contrast-induced nephropathy (CIN) is one of the most important causes of AKI in the clinic, and of interest, the main risk factors (hyperglycaemia, hypertonicity as radiocontrast agents and hypoxia) are very well-established activators of aldose reductase and the polyol pathway. One of the main activators of aldose reductase is hypertonicity. In this regard, radiocontrast agents vary significantly in its osmolarity, with agents that are markedly hyperosmolar (1,500 to 1,800 mOsm kg$^{-1}$, high contrast agents), moderately

hyperosmolar (600–800 mOsm kg$^{-1}$, termed low osmolarity agents, such as Isovue) or iso-osmolar (290 mOsm kg$^{-1}$, such as iodixanol)[33]. Today the most commonly administered agents are low osmolarity agents, which, however, remain hyperosmolar compared with plasma. While there is some controversy over the relative toxicity of radiocontrast agents in relation to their osmolarity[34], most studies suggest that the risk for AKI is greater with high osmolal agents followed by low osmolarity agents and finally iso-osmolar agents[33]. We hypothesize that this is because the higher the osmolarity of the agent, the greater the stimulation of aldose reductase.

To determine the specific deleterious role of endogenous fructose production after aldose reductase activation and its metabolism though fructokinase, we evaluated the effect of radiocontrast agents in diabetic wild type and fructokinase-deficient mice as previously described elsewhere[35,36]. Confirmed diabetic mice at week 2 after streptozocin dosage (Fig. 10a) were then administered either corresponding vehicles or N$^G$-nitro-L-arginine methyl ester (L-NAME), and indomethacin 15 min before the injection of contrast[37]. The low osmolal radiocontrast agent, isovue-370, was then administered to half of the mice with the others serving as controls[37].

As shown in Fig. 10b,c, the administration of radiocontrast with L-NAME/indomethacin resulted in a significant elevation in serum creatinine and urinary NGAL in wild-type animals that was significantly reduced in the diabetic fructokinase-deficient mice. Figure 10d shows the renal histology obtained at 24 h in these mice. While diabetic wild-type diabetic and fructokinase-deficient mice that did not receive radiocontrast showed normal renal histology, severe renal injury with loss of proximal tubular brush border and with tubular vacuolization was observed in wild type but not fructokinase-deficient mice administered radiocontrast. This is similar to the histologic changes of

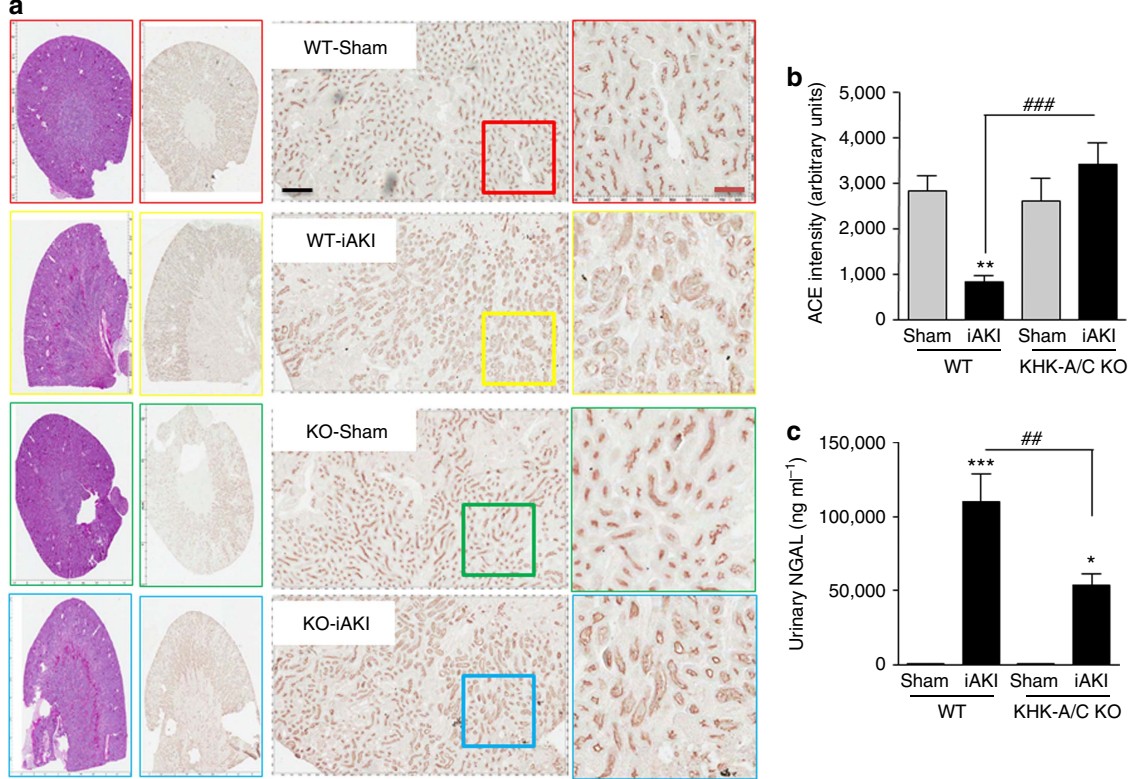

**Figure 5 | Reduced brush border loss and NGAL excretion in fructokinase knockout mice undergoing iAKI.** (**a**) Representative images of ACE staining—brush border marker—of sham operation and mice (wild type (WT) or fructokinase deficient (KHK-A/C KO) undergoing iAKI). (**b**) Intensity quantitation of ACE staining in the same groups of animals. (**c**) Urinary NGAL levels in the same groups of animals. Data indicate mean ± s.e.m. *$P < 0.05$, **$P < 0.01$ and ***$P < 0.001$ versus baseline or within the same group. $n = 5$–9 animals per group with two different studies ###$P < 0.001$, ##$P < 0.01$. Statistical analysis was performed with ANOVA with *ad hoc* analysis by Bonferroni's method to compare all columns. Scale bar: black 500 μm, red 100 μm.

radiocontrast-induced AKI in humans[37–39]. Furthermore, increased renal sorbitol and fructose concentrations in contrast-receiving mice strobgly suggests that the deleterious role of contrast in inducing AKI in mice is mediated by aldose reductase and the activation of the polyol pathway (Fig. 10e,f).

## Discussion
Recently, the National Institute of Diabetes and Digestive and Kidney Diseases sponsored the 'Clinical Trials in Acute Kidney Injury: Current Opportunities and Barriers' workshop to address the need of developing novel therapeutic agents based on precise understanding of key pathophysiological events and the implementation of well-designed clinical trials[40]. In this manuscript, we identify a new, yet unrecognized target, the polyol pathway and fructokinase, whose blockade could contribute to the prevention of iAKI and to accelerate kidney recovery post-insult.

We and others have already postulated an important deleterious role for fructose metabolism in the progression of kidney disease[6,41]. Fructose is a key constituent of table sugar (accounting for 50%) and high-fructose corn syrup (55–65%), and sugar consumption has been shown to be associated with chronic kidney disease in humans[42]. Furthermore, increased sugar consumption has been recently considered as an important risk factor in the current epidemic of non-traditional Mesoamerican chronic kidney disease[43,44]. Thus, these studies suggest that targeting fructose metabolism and limiting sugar consumption may be of benefit to ameliorate chronic kidney disease.

Importantly, besides the dietary source, fructose can be endogenously generated in the kidney cortex by the activation of aldose reductase and the polyol pathway. We have previously

shown that the metabolism of endogenous fructose by fructokinase is an important deleterious step in the progression of diabetic kidney disease[6]; however, to date no studies have characterized the importance of this pathway or the role of fructokinase in AKI despite the observation that fructose can cause acute tubulointerstitial injury in animal models[41]. Aldose reductase is a rate-limiting enzyme which expression is minimal in the kidney cortex under normal conditions. However, upon stimulation, aldose reductase expression is upregulated and the polyol pathway activated, thus converting glucose to sorbitol and then to fructose by sorbitol dehydrogenase, an enzyme constitutively expressed in the kidney cortex[23,45]. To date, aldose reductase expression is known to be upregulated by three major mechanisms: hypertonicity as it occurs in the inner medulla and papilla of the kidney or in states of dehydration[9,46], hyperglycaemia as it occurs in diabetes[47,48] and hypoxia[49,50] as it occurs in ischaemic events. We have previously shown that this pathway protects kidney injury induced by heat and dehydration[44] and here, we show that in ischaemic AKI, there is a significant upregulation of aldose reductase that triggers the generation of fructose in a time-dependent manner. Here we show that patients undergoing AKI demonstrate significantly greater urinary fructose excretion indicative of the polyol pathway activation. It is important to note that while we expect that the main site of this activation occurs in the proximal tubule of the kidney, the polyol pathway can be also activated in peripheral tissues including liver and intestine that can significantly contribute to the overall increase in fructose excretion. Furthermore, we show that if we block the metabolism of this endogenously produced fructose in fructokinase-deficient mice, we can prevent renal injury, oxidative stress and dysfunction

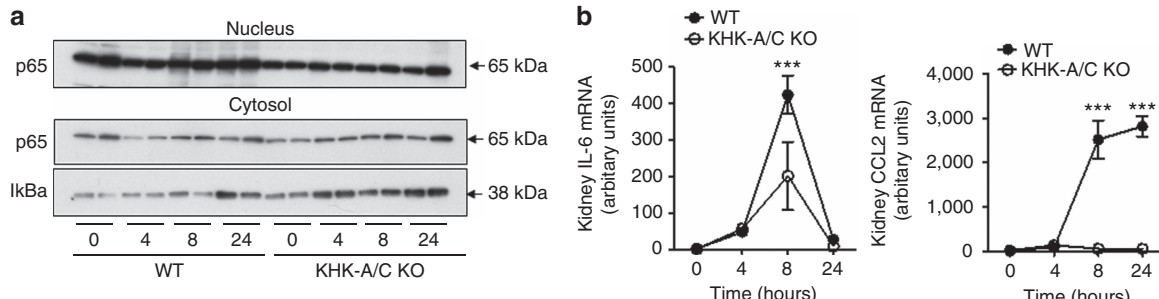

**Figure 6 | Reduced renal oxidative stress in fructokinase knockout mice undergoing iAKI.** (**a**) Time-course analysis of cortical ATP levels in wild type (WT) and fructokinase-deficient (KHK-A/C KO) mice undergoing iAKI for 24 h. (**b**) Time-course analysis of cortical ADP levels in wild type (WT) and fructokinase-deficient (KHK-A/C KO) mice undergoing iAKI for 24 h. (**c**) ADP/ATP ratio in the same conditions. (**d**) Cortical uric acid levels at 24 h in the same groups of animals. (**e**) Top: Representative images of the oxidative stress marker, dihydroethidium (DHE) at baseline and 24 h post iAKI in wild type (WT) and fructokinase-deficient (KHK-A/C KO) mice. Bottom: Representative quantitation of DHE in the same groups of animals. Scale bar: 100 µm. (**f**) Cortical thiobarbituric acid reactive substance levels at 24 h in the same groups of animals. $n = 5$–9 animals per group with two different studies. Data indicate mean ± s.e.m. For (**a**–**c**): *$P < 0.05$, **$P < 0.01$ and ***$P < 0.001$ between WT and KHK-A/C KO. For (**d**–**f**): **$P < 0.01$ and ***$P < 0.001$ versus baseline or within the same group. ##$P < 0.01$. Statistical analysis was performed with ANOVA with *ad hoc* analysis by Bonferroni's method to compare all columns.

**Figure 7 | Reduced renal inflammation in fructokinase knockout mice undergoing iAKI.** (**a**) Representative western blot demonstrating nuclear and cytosolic expression of the NF-κB subunit p65 and their regulator IkBa (uncropped blots are presented as shown in Supplementary Fig. 2). (**b**) Time-course analysis of mRNA levels of the proinflammatory cytokine *il-6* and the chemokine *ccl2* in wild type (WT) and fructokinase-deficient (KHK-A/C KO) mice undergoing iAKI. Data indicate mean ± s.e.m. ***$P < 0.001$ between WT and KHK-A/C KO. $n = 5$–9 animals per group with two different studies. Statistical analysis was performed with ANOVA with *ad hoc* analysis by Bonferroni's method to compare all columns.

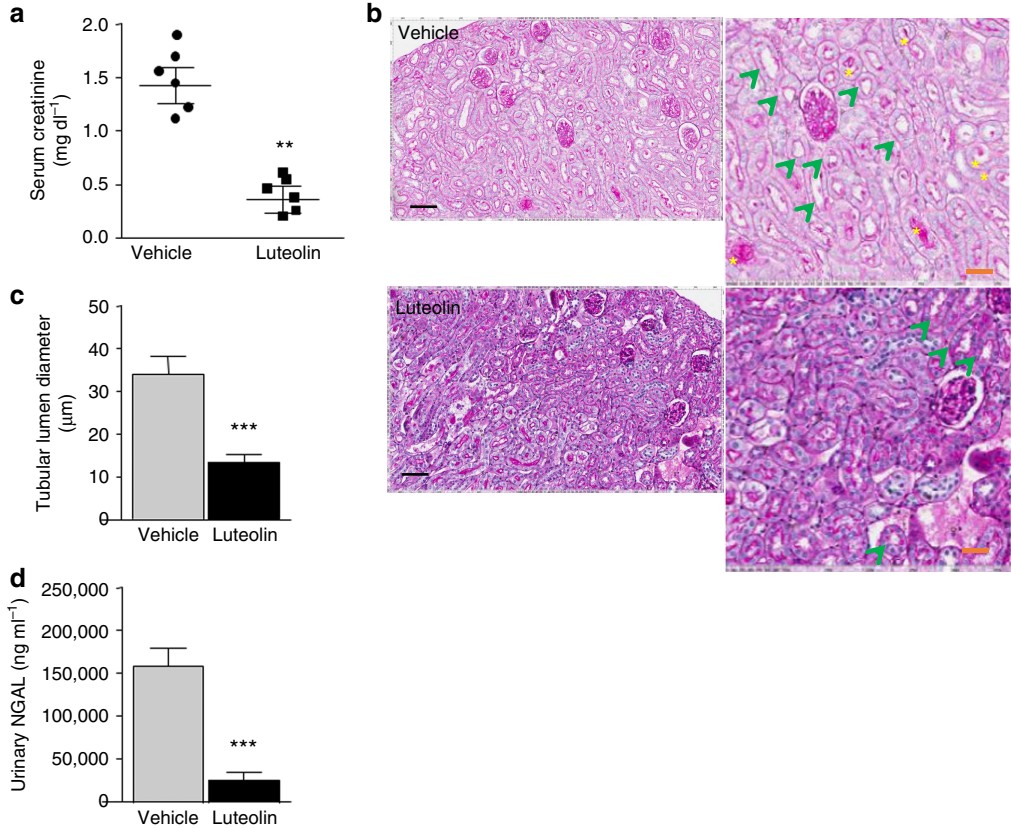

**Figure 8 | Improved renal dysfunction in luteolin-receiving mice before ischaemic insult.** (**a**) Serum creatinine levels in vehicle and luteolin-treated wild-type mice at 24 h post-ischaemic insult. (**b**) Representative periodic acid-Schiff-stained images in the same groups as in **a**. Scale bar: black 300 μm, red 100 μm. (**c**) Tubular lumen area in the same groups as in **a**. (**d**) Urinary NGAL levels in the same groups as in **a**. Data indicate mean ± s.e.m. **\*\*P < 0.01 and \*\*\*P < 0.001. n = 6 animals per group. Statistical analysis by two-tail t-test. Green arrows denote tubules with significant dilatation. Yellow asterisks: cast formation in tubes.

associated with iAKI. In this regard, we have found that luteolin, a potent fructokinase inhibitor, demonstrated significant protection against iAKI in wild-type mice consistent with a previous report in a different model of AKI[31]. Furthermore, we have recently shown new active compounds with efficient fructokinase inhibitory activity[32]. Of these, osthole, a derivative obtained from plants of the Angelica family, demonstrated one of the greatest inhibitory activity and recently, Zheng *et al.*[51] demonstrated its efficiency in ameliorating renal injury in mice undergoing ischaemic AKI. While the specific mechanism whereby osthole exerted protection in AKI was not proposed, we expect based on our data that it could be at least partially mediated by fructokinase inhibition in the renal cortex. Of interest, in our lab, we have tested the efficacy of alternative fructokinase inhibitors with similar strengths obtaining similar protection against iAKI in mice. During the course of our studies, Mirtschink *et al.*[52] reported that ischaemia reperfusion of the heart is also mediated by fructokinase, due to conversion of fructokinase A isoform in cardiac tissues to fructokinase C via a HIF-1α-dependent process. Our studies confirm a role for fructokinase in two models of AKI, namely ischaemia reperfusion and contrast-induced nephropathy, which combined provide a mechanism (ischaemia and upregulation of aldose reductase) for the generation of the substrate necessary for the ischaemic injury. Together, these studies confirm an increasing understanding for the important role for endogenous fructose metabolism in human disease.

Our data would imply that aldose reductase-deficient mice should be protected against AKI as well as fructokinase-deficient

animals as these animals would not produce fructose endogenously. Also blockade of the polyol pathway may exert further benefits over fructokinase inhibition, as it would be associated with reduced sorbitol accumulation, nicotinamide adenine dinucleotide phosphate (NADPH) depletion and the pseudohypoxic effect secondary to NADH accumulation. Consistently, we have previously shown that these mice are protected against diabetic nephropathy, a model that concurs with the activation of aldose reductase in the kidney cortex and the endogenous production and metabolism of fructose[6]. However, since aldose reductase-deficient mice develop polyuria due to their inability to produce sorbitol for a proper urinary concentrating mechanism, it would suggest that fructokinase would be a better target in AKI than the blockade of aldose reductase. The mechanism whereby fructose metabolism exerts toxicity in the proximal tubule seems to be mediated by the generation of local intracellular uric acid as shown in previous reports[20]. Consistently, high uric acid has been commonly considered an important risk factor for AKI and kidney disease in humans[53,54]. Here, we show that in iAKI there is a time-dependent elevation of renal uric acid levels that parallels the activation of the polyol pathway and fructokinase. The elevation in uric acid in wild type but not fructokinase-deficient mice also correlates with an inflammatory response characterized by the nuclear translocation of NF-κB, a well-known transcription factor associated with inflammation in AKI[55-57] and the production of pro-inflammatory cytokines and chemokines. However, despite observing an association between fructose metabolism and uric acid generation, further studies are warranted to establish

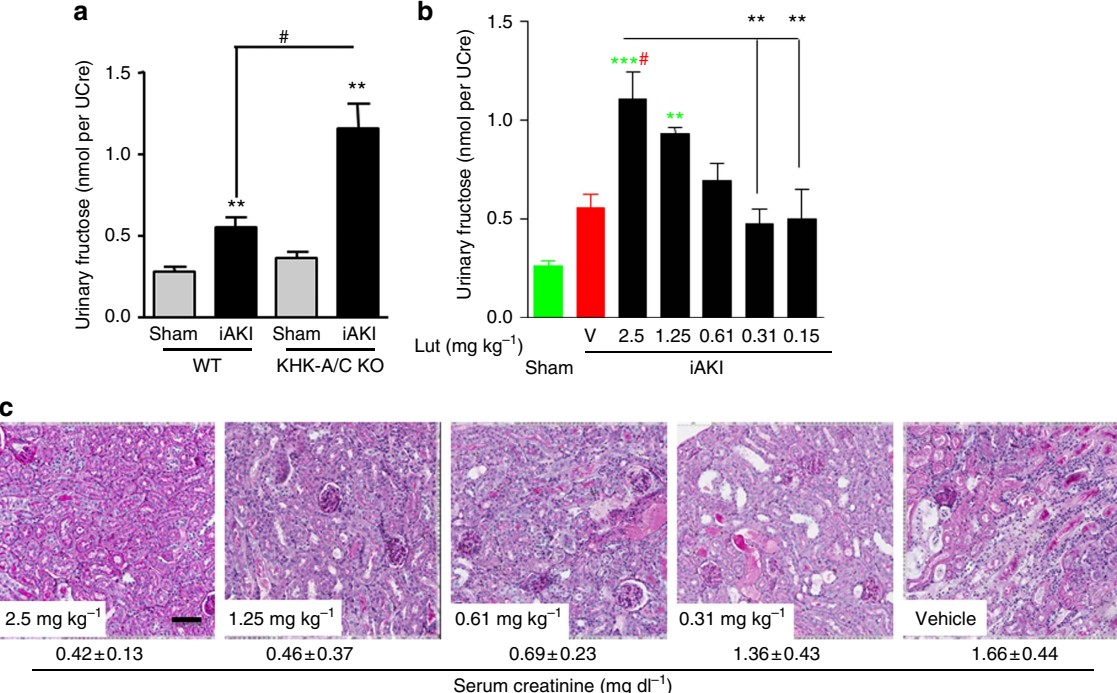

**Figure 9 | Improved renal dysfunction in luteolin-receiving mice after ischaemic insult.** (**a**) Urinary fructose—normalized to creatinine—levels in wild type (WT) and fructokinase-deficient (KHK-A/C KO) mice at 24 h post sham operation or ischaemic acute kidney injury (iAKI). (**b**) Dose–response effect of vehicle (V) or luteolin (Lut) in 24 h urinary fructose production in mice undergoing iAKI. (**c**) Representative periodic acid-Schiff-stained images of the same groups as in (**b**) (below indicate average ± s.d. of serum creatinine obtained for each group). Data indicate mean ± s.e.m. **$P < 0.01$ and ***$P < 0.001$ versus baseline or within the same group. #$P < 0.05$. Statistical analysis was performed with ANOVA with *ad hoc* analysis by Bonferroni's method to compare all columns. Green colour denotes statistical difference versus sham. Red colour denotes statistical difference versus vehicle. $n = 3$ animals per dose. Scale bar: 100 µm.

whether the toxicity induced by fructose metabolism is mediated by uric acid. These studies should include blockade of xanthine oxidase expression in the renal cortex or its activity with allopurinol. In this regard, allopurinol has been positively associated with the amelioration of AKI induced by radiocontrast agents in humans[58,59]. Since uric acid can be generated independently of fructose metabolism due to the ischaemic event, a combination therapy including both of fructokinase and xanthine oxidase inhibitors may exert further protection against iAKI. Another important factor in the described molecular mechanism underlying the pathogenesis of AKI relates to the observed ATP depletion. Our data indicate that ATP depletion in AKI has two components, an early one within the first 2 h post insult in which ATP is depleted from the ischaemia and a later one as a result of fructokinase activation. Consistently, fructokinase-deficient mice despite having similar initial ATP depletion in their kidneys, they are able to restore ATP levels quicker than wild-type animals (Fig. 6a) and therefore demonstrate reduce overall ATP depletion and uric acid generation with less inflammation and oxidative stress. We have proposed before that lowering uric acid could be clinically relevant not only to protect individuals from AKI (as it would occur in renal replacement) but even to treat it. The identification of fructokinase as a late response protein in maintaining reduced ATP levels in the kidney and producing uric acid provides as well with a therapeutic window (first 8–12 h post insult) in which specific fructokinase inhibitors could be used for the treatment of ischaemic AKI (Figs 2 and 9). Furthermore, AMPD2, the renal AMP deaminase isoform, would be a key target in the pathogenesis of AKI as its blockade would prevent both the early and late ATP depletion, and therefore it would be more beneficial to accelerate kidney disease post-AKI.

In summary, although it is important to note that our study is limited to the characterization of the early changes (first 24 h) that occur after ischaemic insult thus not being able to determine potential protective long-term effects of fructokinase blockade in kidney disease (i.e., fibrosis, renal dysfunction, aging-associated kidney disease, etc), our data indicate that in ischaemic AKI, there is an important generation and metabolism of fructose in the kidney cortex that correlates with the development of renal injury and dysfunction. Therefore, fructokinase targeting alone or in combination with therapies toward the reduction of uric acid generation in the kidney may be an important therapeutic approach for the prevention of kidney disease or the acceleration of kidney recovery.

## Methods

**Human samples.** Urinary fructose was analysed from samples collected for a previous study with the approval of both the Colorado Institutional Review Board and the Clinical and Translational Research Center[21]. Written informed consent was obtained for all patients enrolled in the study prior to any sample collection. For this study, samples could not be individually identified by the investigators. Therefore, the use of these samples in this work did not meet the definition of human subject research under federal regulations at 45 CFR 46. Subjects were excluded if they had known underlying chronic kidney disease (preoperative estimated Schwartz clearance $< 80$ ml/min/1.73 m$^2$), exposure to nephrotoxins within 1 week of surgery (intravenous contrast, aminoglycosides), proteinuria (dipstick 1 + or greater), urinary tract infection, diabetes, baseline serum creatinine that was unavailable or inability to obtain consent. The primary outcome assessed was the development of AKI post-cardiac bypass as detailed[21]. AKI was defined, according to RIFLE criteria R, as a 50% or greater increase in serum creatinine at 24 h. Controls refer to subjects with lower than 50% increase in serum creatinine at 24 h. Fresh urine was collected, samples were centrifuged for 5 min and the supernatant was aliquoted and immediately placed in $-80\,°C$ freezer until analysis. Dietary fructose prior to the intervention was not determined and therefore its contribution to the development and progression of AKI could not be assessed.

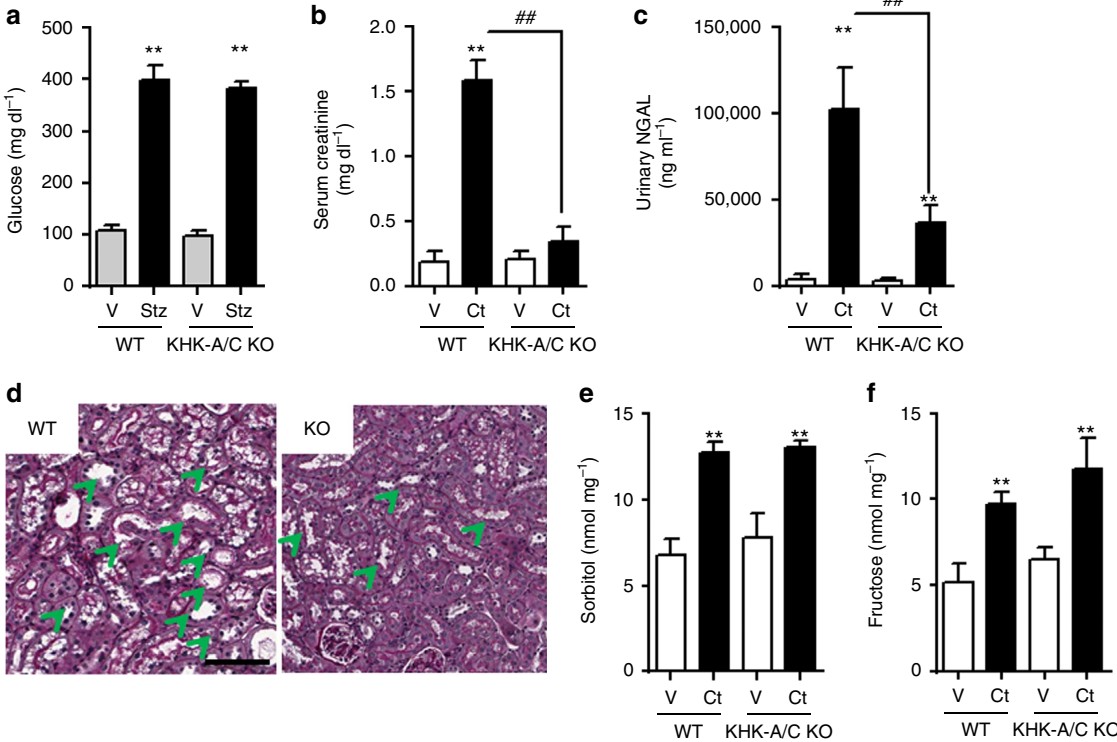

**Figure 10 | Ameliorated renal dysfunction and injury in fructokinase knockout mice undergoing CIN.** (**a**) Serum glucose levels in control and streptozocin injected wild type (WT) and fructokinase-deficient (KHK-A/C KO) mice at 2 weeks post injection. (**b**) Serum creatinine, (**c**) urinary NGAL and (**d**) histology/ periodic acid-Schiff staining. Scale bar: 100 μm. (**d**) Renal cortical sorbitol and (**e**) renal cortical fructose levels in 2-week diabetic animals receiving either vehicle (V) or contrast (Ct) for 24 h. n = 5–9 animals per group. Data indicate mean ± s.e.m. **$P < 0.01$ versus baseline or within the same group. ##$P < 0.01$. Statistical analysis was performed with ANOVA with *ad hoc* analysis by Bonferroni's method to compare all columns. Green arrows denote tubules with significant dilatation.

**Mice.** Global *khk*[−/−] and wild-type controls in the C57/Bl6 background were generated and bred as described previously[60,61]. All mice were bred in the specific pathogen-free barrier facility at the University of Colorado Denver. All animal experiments were performed in accordance with the Animal Care and Use Committee of the University of Colorado. In all experiments described, littermates male mice aged 6–8 weeks with a body weight of 22–25 g were used. In order to avoid variability between mice, temperature and day, all animal procedures for each experiment (i.e., wild type versus knockout, administration of inhibitors, titrations, etc.) were performed the same day. iAKI in tribromoethanol (Avertin) anaesthesized mice was induced by clamping both renal pedicles for 22 min—sham operation consisted in exposing but not clamping the renal pedicles for the same time[29]. In this regard, while areas at sea level require 30 or more minutes of ischaemia to induce significant renal damage[62], our studies being performed at high altitude (5,280 feet over sea level) required shorter ischaemic time (22 min) to induce a similar degree of renal injury and elevation of serum creatinine[29,63,64]. All mice were killed within the first 24 h post-ischaemic insult or sham operation.

CIN was induced in mice as described elsewhere[36,37]. Briefly, streptozocin-dependent 2-week diabetic mice with fasting blood glucose around 375 mg dl[−1] were subjected to either vehicles or $N^G$-nitro-L-arginine methyl ester (L-NAME), 10 mg kg[−1], dissolved in 0.9% saline and indomethacin (indomethacin, 10 mg kg[−1], dissolved in dimethylsulfoxide) to block nitric oxide and prostaglandin synthesis beginning 15 min before the injection of contrast[37]. The low osmolal radiocontrast agent, isovue-370 (iopamidol containing 370 mg iodine per ml, 3 g iodine per kg; Amersham), was then administered to half of the mice with the others serving as controls[37]. Renal injury and function was determined 24 h after injection of radiocontrast agents.

**Laboratory studies.** Serum and urine creatinine were measured with Albuwell M (Exocell, Philadelphia, PA) and Creatinine LiquiColor Test (Enzymatic Methodology; Stanbio, Boerne, TX), respectively. Urine levels of NGAL were measured with a Mouse NGAL ELISA kit (R&D Systems, Minneapolis, MN). BUN was determined with an VetACe autoanalyzer. Urinary fructose was determined biochemically (EFRU-250; Bioassay Systems) following the manufacturer's instructions. This kit employs fructose dehydrogenase to initiate the reaction, a plant enzyme not found in mammals thus not interfering with activities of endogenously present mammalian fructokinase.

**Histological analysis.** Formalin-fixed, paraffin-embedded sections (2.5 μm) were stained with the periodic acid-Schiff reagent for light microscopy. Kidney sections were scanned using an Aperio Scanscope and observed by two investigators in a blinded manner. On coronal sections of the kidney, tubules were analysed for tubular dilatation and tubular lumen area as described[6].

**Determination of AR activation in kidney cortex.** Kidney cortex were homogenized in a cold buffer containing 150 mM KCl, 20 mM Tris (pH 7.5), 1 mM EDTA (pH 8) and 1 mM DTT. The protein content of the lysate was determined using Pierce BCA protein assay (Thermo Fisher Scientific, Rockford, IL). Aldose reductase activity was determined as follows: 50 μg of lysate were incubated with 50 mM potassium phosphate buffer pH 6.2, 0.4 M lithium sulfate, 5 mM 2-mercaptoethanol, 10 mM DL-glyceraldehyde, 0.1 mM NADPH oxidase. The assay mixture was incubated at 37 °C and initiated by the addition of NADPH, and AR activity was determined by measuring the change in the absorbance at 340 nm with a plate reader (Biotek Synergy 2) and expressed as the μmol of DL-glyceraldehyde consumed per hour. Appropriate blanks were used to subtract the natural decay of NADPH in the samples. Cortical and urinary sorbitol, fructose and uric acid levels were determined in the same lysates using fluorometric kits (sorbitol: K631-100 Biovision, fructose: enzychrom fructose kit, bioassays, uric acid' enzychrom uric acid assay, bioassays) as per the manufacturer's protocol.

**Immunohistochemistry.** Methyl Carnoy's solution-fixed, paraffin-embedded sections were used for immunohistochemistry[65]. A rabbit anti- ACE antibody (Chemicon) was used as the primary antibody. Briefly, after deparaffinization, the sections were treated with 3% $H_2O_2$ for 10 min to inactivate endogenous peroxidase activity. After incubation with a background sniper (Biocare Medical, Concord, CA) for 15 min, sections were incubated with primary antibodies overnight at 4 °C. The sections were also incubated with rabbit anti-IgG secondary antibodies for 30 min before immunoperoxidase staining was conducted using the Mach2 rabbit HRP polymer (Biocare Medical). To assess the ACE-positive area, the digital images at ×400 magnification were analysed using Image scope software (Aperio Technologies, Vista, CA). The percent positive area was determined as the 3,3-diaminobenzidine-positive pixel values per examined interest area in each section.

**Immunofluorescence on mice kidneys.** Immunofluorescence against aldose reductase and ACE was performed on 10% goat serum in phosphate-buffered saline blocked kidney slices further incubated overnight with primary antibodies (AR antibody, kind gift from Dr Mark Petrash and ACE antibody (AF1513, R&D)). The following day, slides are rinsed and incubated with Alexa Fluor-conjugated secondary antibody (Molecular Probes) against the specific IgG of the primary antibody. For nucleus identification, DAPI staining is used in combination with a fluorescence fading retardant (Vector Laboratories, Burmingdale, CA) before imaging by confocal microscopy. Immunostained preparations were imaged and analysed using a laser-scanning confocal microscope (LSM510, Carl Zeiss, Thornwood, NY) with a × 40 water immersion objective and the corresponding postacquisition software. Immunofluorescence analysis was performed from sections from three animals and by evaluation of >10 random fields each[66].

Renal oxidative stress by dihydroethidium staining was determined following the manufacturer's protocol (D1168, Thermo).

**Western blotting.** For aldose reductase and fructokinase expression (Fig. 2), protein lysates were obtained after homogenization of renal cortical tissues (50 mg) in MAPK lysis buffer[67] containing 0.5% triton X-100, 50 mM β-glycerophosphate, 2 mM $MgCl_2$, 1 mM EGTA, 1 mM DTT and a cocktail of protease inhibitors (Roche). Homogenates were centrifuged in cold at 13,000 r.p.m. for 10 min and supernatants collected for protein quantification using the BCA assay (Pierce). Nuclear and cytosolic extracts were extracted from the kidney cortex using the nuclear/cytosol fractionation kit from Biovision. Pure nuclear and cytosolic fraction were determined by analysing the expression of specific markers (Lamin A/C for nucleus and tubulin for cytosol). Western blot was performed using specific primary antibodies at a concentration of 1:1,000 in TTBS as follows: aldose reductase (kind gift from Dr Mark Petrash at University of Colorado), fructokinase (Sigma, HPA007040), p65 and IκBα (Cell signaling, 8242 and 4814) and detected with HRP-conjugated antibodies (Cell Signaling) and Immun-star (Bio-Rad).

**Real-time PCR.** Cytosolic RNA was isolated from mice kidney using the RNeasy kit (Qiagen, Valencia, CA). Before real-time PCR, RNA was converted to cDNA using the iScript reverse transcriptase kit (Bio-Rad) as described by the manufacturer. RT-PCR primers specific to IL-6: 5′-ACCGCTATGAAGTTCCT CTC-3′ (F), 5′-CCTCTGTGAAGTCTCCTCTC-3′(R); CCL2: 5′-GAAGGAATGG GTCCAGACAT-3′ (F), 5′-ACGGGTCAACTTCACATTCA-3′ (R); and β-actin: 5′-CGTGCGTGACATCAAAGAG-3′ (F), 5′-TGCCACAGGGATTCCATAC-3′ (R) were designed using Beacon Designer 5.0 software (Premier Biosoft International, Palo Alto, CA). RT-PCR was performed using 70 nM primers and the SYBR Green JumpStart Taq Readymix QPCR kit (Sigma) on a Bio-Rad I-Cycler. RT-PCR runs were analysed by agarose gel electrophoresis and melt curve to verify that the correct amplicon was produced. β-Actin RNA was used as an internal control, and the amount of RNA was calculated by the comparative $C_T$ method as recommended by the manufacturer[29].

**Cell culture.** Immortalized human proximal tubule cells (HK-2) were obtained from the ATCC (CRL-2190) and cultured in keratinocyte-SFM medium (Invitrogen) supplemented with 10% FBS, penicillin (100 U $l^{-1}$), streptomycin (100 μg $l^{-1}$), human recombinant EGF and bovine pituitary extract. Cells were cultured at 37 °C to 60–70% confluency and then harvested with a non-denaturing cold buffer containing 150 mM KCl, 20 mM Tris (pH 7.5), 1 mM EDTA (pH 8) and 1 mM DTT. The protein content of the lysate was determined using Pierce BCA protein assay (Thermo Fisher Scientific Inc., Rockford, IL) for fructokinase activity analysis in the presence of luteolin.

**Fructokinase activity in HK-2 cells.** Fructokinase activity in HK-2 lysates was determined by measuring the amount of fructose-dependent ATP depletion in a kinetic reaction as follow. As control, HK-2 cells deficient for KHK expression were employed to calculate background ATP depletion during the length of the assay. The ATP depletion cell lysate reaction contained 100 μg of lysate protein, 50 mM imidazole, 4 mM magnesium chloride, 1 M potassium acetate (pH 7.5), 20 mM n-acetylglucosamine, 40 mM sodium fluoride, 5 mM fructose and 5 mM ATP in a total reaction volume of 200 μl. Lysates were preincubated with different concentrations of luteolin in the reaction mixture for 15 min at 37 °C. After adding in the fructose (except for negative controls), the reaction was continuously shaken and incubated at 37 °C for 120 min. Samples of each reaction were taken after 2 min and 120 min for measuring ATP levels. ATP levels were measured using an ATP Assay Kit (BioVision, Inc., Milpitas, CA). Five microlitres of sample was added to 95 ul of the ATP reaction mixture and readings were taken at 570 nm. The change in ATP levels were calculated and the percentage ATP inhibition was determined. $IC_{50}$s for inhibiting ATP depletion was calculated from the dose response (0.1–500 μM).

**Statistical analysis.** All values are expressed as means ± s.e. Statistical analysis was performed with either ANOVA using Bonferroni's method to compare the groups or by two-tail t-test analysis as indicated in the figure legends. A level of $P < 0.05$ was considered statistically significant.

**Data availability.** The data sets generated during and/or analysed during the current study are available from the corresponding author on reasonable request.

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

## Acknowledgements

This work has been supported by NIH grants 5K01DK095930 and R03DK105041to M.A.L. as well as a K supplement fund from the School of Medicine at the University of Colorado.

## Author contributions

A.A.-H., W.C., T.N., R.J.J. and M.A.L. designed the research; A.A.-H., N.L., C.C., S.I., W.C., C.R.-J., T.M., T.I and M.A.L. performed the research; M.T.L., M.W., M.F. and M.A.L. contributed to new reagents/analytic tools; A.A.-H., W.C., S.I., R.J.J. and M.A.L. analysed data; A.A.-H., R.J.J. and M.A.L. wrote the paper.

## Additional information

**Competing financial interests:** R.J.J. has grants with Amway and Danone and is on the Scientific Advisory Board of Amway. R.J.J. also has authored two lay books on sugar (fructose) and its role in obesity and metabolic syndrome (The Sugar Fix, Rodale, 2008 and The Fat Switch, mercola.com, 2012). M.A.L., T.I and R.J.J. have patents and patent application related to the blockade of fructokinase in the prevention and treatment of sugar-induced metabolic syndrome and renal disease. M.A.L., M.T.L., C.R.-J., M.F.W. and R.J.J. are members of Colorado Research Partners, LLC, a company focused on the development of fructokinase inhibitors. All other authors declare no competing financial interests.

