## [Peer review file · Nature Communications]

Reviewers' comments:

Reviewer #1

Remarks to the Author:

A very interesting set of studies designed to delineate the role of FK in the pathophysiology of ischemic AKI. There is a logical progression from polyol pathway activation to use of FK deficient mice and a reduction in ischemic injury. The use of luteolin was also consistent with the hypothesis. This is of high interest to the community as it may offer another pathway to minimized ischemic AKI and enhance the rate of recovery. However, necessary controls are lacking and too often the data do not allow the interpretation offered. This is especially true for the histology presented. Of particular note:

1. The authors assume urinary fructose post ischemic injury secondary to cardiac surgery is from kidney production. It is more likely from filtration of fructose and lack of PT uptake. This needs to be considered as an alternate pathway for urine fructose.
2. Were the genetic backgrounds of the fructokinase deficient mice the same as the wild type mice? This is important as different strains have different sensitivity to ischemic injury.
3. Are you sure the AR staining in figure 2 is primarily PT in origin? This interstitial space seems to be the source of labeling in many cases, although it does appear PT are also labeled.
4. The histology in figures 4, 7 and 8 are difficult to evaluate due to magnification and focus. I would suggest a higher power and thinner sections to make the point you are aiming for.

Reviewer #2

Remarks to the Author:

In this manuscript wild type and fructokinase-deficient mice underwent ischemic AKI or sham operation and renal function and injury was assessed at 24 hours after ischemia. Ischemic animals had an increase in serum creatinine, BUN and urinary NGAL as well as uric acid and ATP depletion. Fructokinase-deficient mice were protected. In addition, a fruktokinase inhibitor exerted protection, and the authors conclude that it accelerated kidney recovery in wild-type mice. The authors present the data as evidence of support for the polyol pathway and generation of fructose as an important mediator for ischemic injury.

1. While the results are interesting, it adds to a large number of studies demonstrating that a particular intervention or knockout animal is protected against acute kidney injury without sufficient in-depth mechanistic exploration that would raise hope for translation of the observation to humans.
2. Figure 1: the authors present urinary fructose as nmol/Ucre. They conclude that this means that there has been increased kidney fructose produced. Perhaps I am overlooking it, but I do not see that the authors have described the patients with AKI. This is important for a number of reasons. One of the reasons is that the nmol/Ucre may not reflect an increase in kidney production. This can

occur because there is a marked decrease in urinary creatinine excretion during the non-steady state conditions of acute kidney injury. A decrease in urinary creatinine per se could result in an increase in the urinary fructose nmol/Ucre ratio without any increase in total kidney production of fructose. Furthermore, it would be important to know that there is no fructokinase activity in the urine.

3. The authors have used one condition of ischemia reperfusion, that is, a 22 minute bilateral clamp. It is somewhat surprising that the creatinine rose to such high levels with this moderate period of clamp. Furthermore, we are only presented with one timepoint at 24 hours. With the increase in creatinine at these levels at 24 hours, it would be important to show that the animals survived the acute insult.

4. In figure 2a, the authors conclude that fructokinase expression is increased post AKI. It is not clear that this is the case if one considers that there seems to be an increase in actin on these gels. A more accurate representation would be to normalize the KHK signal to the actin signal.

5. In one case (figure 2b), the authors determine brush border integrity by phalloidin staining, whereas in another case (figure 5b), they evaluate brush border loss by expression of ACE in the renal cortex. It is not clear why they do not evaluate the ACE and phalloidin both in both situations.

6. The authors have just looked at one model with one timepoint of ischemic period and one timepoint of evaluation at 24 hours. This is inadequate to draw definitive conclusions on the role of this pathway in kidney injury. Are similar findings found with toxin models? What happens to survival of the animals over time? And what about the development of fibrosis?

7. On what compartment is the protective effect manifest most importantly? Tubular? Vascular? Interstitial?

Reviewer #3

Remarks to the Author:

This manuscript hypothesizes that in mouse models of ischemic acute kidney injury (iAKI), endogenous fructose production generated by the polyol pathway is the underlying mechanism causing iAKI. The authors determined levels of standard biomarkers of renal injury in fructokinase-KO (FKKO) they previously used in related work (JASN 2014, Endogenous fructose production and fructokinase activation mediate renal injury in diabetic nephropathy), to evaluate the role of this pathway this time in iAKI, a clinically important problem with limited treatment options.

Major comments:

1. The manuscript is well written by this group that has been very productive for years in evaluating the role of fructose in renal complications. The hypothesis as stated (endogenous ... by the polyol pathway) can be revised to better reflect the results as presented. This FKKO approach though indirect can be validated if use of aldose reductase-KO (ARKO) mice or of available AR inhibitors result in similar outcomes. In fact, an AR inhibitor ameliorates LPS inflammation-induced acute

kidney injury in mice (Takahashi et al 2012 Plos One), supporting their findings.

2. Elimination of endogenous fructose production via AR seems to result in an outcome that may confound findings in FKKO. Ho et al 2000 Mol Cell Biol. 2000 found that AR deficiency in mice caused a partially defective urine-concentrating ability resulting in nephrogenic diabetes insipidus.

3. Fig. 1 is indicated to be due to activation of the polyol pathway, but this claim can be strengthened by data on fructose consumption by patients (the excreted fructose may not be endogenous). Fig. 1 is a ratio of urinary fructose to creatinine in which both the ratio and the creatinine increase in AKI. As such, fructose concentration must be very low, especially if it is reabsorbed by the kidney. However, this manuscript did not state what method was used to determine nmol (Figs. 1, 2, 9) concentrations of fructose.

4. Since FKKO cannot metabolize fructose, and as suggested by the manuscript fructose is continuously being produced by the polyol pathway throughout the body and renal fructose concentrations (excretion) seem very low, where is the fructose going in this mouse model that keeps on making but not eliminating fructose, and is there significant accumulation in the tissue or blood which could affect findings?

5. The finding that the flavonoid luteolin is also a FK inhibitor is exciting and innovative, but this compound has numerous biological effects (reviewed by Lopez-Lazaro 2009 Mini Rev Med Chem). The authors' interesting finding of FK inhibition requires validation of luteolin specificity. Because they are the leading investigators in fructose physiology, their work will likely be cited by other workers and the public may consume commercially available luteolin while partaking of fructose laden foods. The authors should discuss potential side effects of luteolin in other organ systems, and off target effects in the kidney, especially on other related kinases. Since the drug is given intravenously, these indirect effects may confound findings presented in Figs. 8 & 9.

6. It is not clear how 941 nM can statistically be the IC₅₀, as there seems to be only one point between 0 and 100% inhibition (Fig. 8A). What is the standard error for this estimate? What is the rationale for the 2.5 mg/kg in vivo dose?

Minor concern:

1. Since this is a general and not a nephrology journal, readers should be guided accordingly as to where the tubular dilatation and cast formation are in Fig. 4, at least with arrows and directions. Similar layperson explanations should be made in Figs. 5, 6, 8 and 9. Fig. 2 was well presented.

Response to reviewers:

General comment: We appreciate the critiques of the Reviewers and Editors which were very helpful and have led to a markedly improved manuscript. We believe, in the revised version, we have significantly improved the quality of the manuscript and addressed the concerns raised by the reviewers hoping that the paper is now acceptable for publication in the journal. Overall, the manuscript has been well received although serious criticism was raised. In order to fully address the reviewer's and editor's concern we have taken up the majority of the allotted time for revision which has allowed us to examine the potential deleterious role of this pathway in a different model of acute kidney injury (contrast-induced nephropathy) as well as generate new data to better characterize the underlying mechanistic model.

Reviewer #1

A very interesting set of studies designed to delineate the role of FK in the pathophysiology of ischemic AKI. There is a logical progression from polyol pathway activation to use of FK deficient mice and a reduction in ischemic injury. The use of luteolin was also consistent with the hypothesis. This is of high interest to the community as it may offer another pathway to minimized ischemic AKI and enhance the rate of recovery. However, necessary controls are lacking and too often the data do not allow the interpretation offered. This is especially true for the histology presented. Of particular note:

1. The authors assume urinary fructose post ischemic injury secondary to cardiac surgery is from kidney production. It is more likely from filtration of fructose and lack of PT uptake. This needs to be considered as an alternate pathway for urine fructose.

We like to thank the reviewer for considering this work being “logical” and “of high interest”. The reviewer is correct in that we cannot at this point separate whether the observed urinary fructose excretion in subjects undergoing AKI is produced locally in the kidney or elsewhere and accumulated in the urine due to reduced tubular reabsorption. Consistently, our data in this paper as well as our published data¹ demonstrate that fructose causes proximal tubular damage, in particular in the S3 segment where fructokinase and the main fructose transporter GLUT5 is expressed. Therefore, it is logical as the reviewer suggest that elevated urinary fructose is a consequence of reduced reabsorption. It is also likely that the tubular injury result in cell shedding from the proximal tubule –as it occurs with the tubular injury marker NGAL in figures 5 and 8- thus releasing the locally produced fructose and therefore both effects (increased tubular shedding and reduced reabsorption contribute to its elevation in the urine). Therefore in our discussion section and as suggested by the reviewer we have added a paragraph considering that elevated urinary fructose excretion could potentially be due to not only local damage in the tubule and consequent release of locally produced fructose but also due to the inability of its tubular reabsorption.

2. Were the genetic backgrounds of the fructokinase deficient mice the same as the wild type mice? This is important as different strains have different sensitivity to ischemic injury.

Both wild type and fructokinase deficient mice shared the same genetic background (C57BL6). In order to reduce variability due to genetic differences between animals, all mice in the study were littermates obtained by breeding mice heterozygous for fructokinase. This way we could obtain both wild type and fructokinase knockout littermates. A paragraph indicating equal background and that we worked with littermates to reduce variability has been included in the methods section.

3. Are you sure the AR staining in figure 2 is primarily PT in origin? This interstitial space seems to be the source of labeling in many cases, although it does appear PT are also labeled.

As explained in the methods section, in order to ensure that AR staining was limited to the proximal tubule, images were analyzed by confocal microscopy. Usage of confocal microscopy allowed us to not only obtain clear fields for colocalization studies but also to delimitate the area of the proximal tubule and quantify the fluorescence signal in that area of interest removing potential background signal coming from the interstitium. Also, it is important to note that the AR (akr1b3) antibody employed was custom made and provided by Dr Mark Petrash, a renowned expert in the characterization of aldose reductase and the polyol pathway, and tested by us before in AR deficient mice².

4. The histology in figures 4,7 and 8 are difficult to evaluate due to magnification and focus. I would suggest a higher power and thinner sections to make the point you are aiming for.

We agree with the reviewer and therefore we have now added new images with higher magnification to figures 4, and 8 –no histology is shown in figure 7-.

Reviewer #2

In this manuscript wild type and fructokinase-deficient mice underwent ischemic AKI or sham operation and renal function and injury was assessed at 24 hours after ischemia. Ischemic animals had an increase in serum creatinine, BUN and urinary NGAL as well as uric acid and ATP depletion. Fructokinase-deficient mice were protected. In addition, a fruktokinase inhibitor exerted protection, and the authors conclude that it accelerated kidney recovery in wild-type mice. The authors present the data as evidence of support for the polyol pathway and generation of fructose as an important mediator for ischemic injury.

1. *While the results are interesting, it adds to a large number of studies demonstrating that a particular intervention or knockout animal is protected against acute kidney injury without sufficient in-depth mechanistic exploration that would raise hope for translation of the observation to humans.*

We thank the reviewer for finding our work interesting. We share the same concern related to the lack of translational success from animal models to human studies. Therefore, in order to increase the clinical relevance of our mouse study, in collaboration with Drs Faubel and Dennen at the University of Colorado, we analyzed urinary fructose excretion in human subjects undergoing AKI. The data presented in figure 1 suggests elevated fructose excretion post AKI which was observed to be total (-as suggested by the reviewer in question 2 below- new figure 1A) and normalized to creatinine (Figure 1B). Although we acknowledge that our study is somewhat preliminary for being translated into human studies, our data in both animal and human subjects suggests that the determination of urinary fructose could be potentially relevant to clinically predict and diagnose undergoing AKI in human subjects. Furthermore, as requested by the Editor, we have expanded our study and have determined the importance of the endogenous fructose production and metabolism in a different model of acute kidney injury, contrast-induced nephropathy (CIN). CIN is one of the most important causes of AKI in the clinic, and of interest, the main risk factors (hyperglycemia, hypertonicity as radiocontrast agents and hypoxia)

are very well established activators of aldose reductase and the polyol pathway. In this new work, supported by a new NIH R03 grant supplementary to the PI's K award, we demonstrate that fructose is endogenously produced in the kidney cortex of mice undergoing CIN and that the blockade of fructokinase is significantly important for the prevention of CIN. We have now included this data that supports that the endogenous fructose production and metabolism is pathway in which other forms of AKI converge into, as figure 10. Furthermore, and in regards to the clinical relevance of our study, our published data in human subjects undergoing Mesoamerican nephropathy, a non-traditional form of chronic kidney disease with a high mortality that economically affects numerous countries in Central America, we postulate that this form of CKD is a consequence of daily episodes of AKI caused by heat and dehydrating conditions that in one hand activate the endogenous fructose production and metabolism and on the other induced uric acid-dependent crystalluria³, based on these observations, our group in collaboration with several organization like La Isla Foundation have started clinical trials in the role of reducing urinary osmolality/hypertonicity, reducing the amount of dietary sugar from beverages, and using specific blockers of fructokinase. In terms of a deeper understanding in the molecular mechanism of AKI, we postulate in this manuscript for the first time that the observed ATP depletion that occurs in AKI has two components, an early one within the first 2 hours post insult in which ATP is depleted from the ischemia and a later one as a result of fructokinase activation. Consistently, fructokinase deficient mice despite having similar initial ATP depletion in their kidneys, they are able to restore ATP levels quicker than wild type animals (figure 6A) and therefore demonstrate reduce overall ATP depletion and uric acid generation with less inflammation and oxidative stress. Consistently, we have proposed before that lowering uric acid could be clinically relevant not only to protect individuals from AKI (as it would occur in renal replacement) but even to treat it. The identification of fructokinase as a late response protein in maintaining reduced ATP levels in the kidney and producing uric acid provides as well with a therapeutic window (first 8-12 hours post insult) in which specific fructokinase inhibitors could be used for the treatment of ischemic AKI (figures 2 and 9). More mechanistically, and if our hypothesis is correct which is based on the data presented in this manuscript, AMPD2, the renal AMP Deaminase isoform, would be a key target in the pathogenesis of AKI as its blockade would prevent both the early and late ATP depletion and therefore it would be more beneficial to accelerate kidney disease post-AKI. The characterization of AMPD2 in AKI is the focus of further studies led by the PI. A paragraph discussing a deeper more mechanistic downstream effects on fructokinase blockade has been added to the discussion as per reviewer's suggestion.

2. Figure 1: the authors present urinary fructose as nmol/Ucre. They conclude that this means that there has been increased kidney fructose produced. Perhaps I am overlooking it, but I do not see that the authors have described the patients with AKI. This is important for a number of reasons. One of the reasons is that the nmol/Ucre may not reflect an increase in kidney production. This can occur because there is a marked decrease in urinary creatinine excretion during the non-steady state conditions of acute kidney injury. A decrease in urinary creatinine per se could result in an increase in the urinary fructose nmol/Ucre ratio without any increase in total kidney production of fructose. Furthermore, it would be important to know that there is no fructokinase activity in the urine.

We thank the reviewer for bringing up this important point. The reviewer is correct in that the fructose data in figure is normalized to urinary creatinine as a means to compensate for overall urinary output. We have updated figure 1 with the overall fructose excretion (in nmol) without normalization. As seen in the figure, urinary fructose is significantly higher in patients undergoing AKI suggestive of increased fructose production locally. As explained above for reviewer 1, we have modified our text to acknowledge the possibility that

increased urinary fructose could be due not necessarily to elevated local production but also to an impairment in its reabsorption by the proximal tubule. We have as well by the reviewer's suggestion, expanded our methods section describing the population used in our AKI model which correspond to kids undergoing cardiac bypass. As per reviewer's suggestion we analyzed fructokinase activity by measuring the ability of fructose to deplete ATP in the urine samples and could not find any significant fructokinase activity. In this regard, a brief comment has been added to the results section.

3. The authors have used one condition of ischemia reperfusion, that is, a 22 minute bilateral clamp. It is somewhat surprising that the creatinine rose to such high levels with this moderate period of clamp. Furthermore, we are only presented with one timepoint at 24 hours. With the increase in creatinine at these levels at 24 hours, it would be important to show that the animals survived the acute insult.

The raise in serum creatinine after bilateral clamping has been fully characterized over the last 10-15 years⁴⁻⁶. In this regard, it is very well established by us and others that times varying from 20-26 minutes are enough to significantly raise serum creatinine to levels above 1.25 mg/dl in mice. The reasoning whereby researchers choose 22 or 26 minutes rely on multiple factors, including the altitude. While experiments run at sea require longer periods of time to significantly to significantly induce renal injury, at high altitude (like Denver, CO at 5280 feet above sea level), a time of 22 minute clamping is sufficient to significantly raise serum creatinine. Nevertheless, in studies led by Dr Andres-Hernando as part of Dr Faubel lab, she demonstrated that animals survive the acute injury for over 7 and even 28 days. A brief description in the methods section justifying employment of 22 minute clamping has been added to the main manuscript.

4. In figure 2a, the authors conclude that fructokinase expression is increased post AKI. It is not clear that this is the case if one considers that there seems to be an increase in actin on these gels. A more accurate representation would be to normalize the KHK signal to the actin signal.

We agree with the reviewer and therefore we have calculated intensity densitometry of the b lots for AR, KHK and Actin, and graphed the densitometry for AR and KHK normalized to actin expression. As shown in new figure 2, AR expression remains significantly higher at 8 and 24 hours post-insult after actin normalization.

5. In one case (figure 2b), the authors determine brush border integrity by phalloidin staining, whereas in another case (figure 5b), they evaluate brush border loss by expression of ACE in the renal cortex. It is not clear why they do not evaluate the ACE and phalloidin both in both situations.

We apologize for this mistake. In figure 2b as it was shown in the figure, figure legend and the methods section we correlated AR with ACE expression but not phalloidin. We have not used phalloidin in either figure 2b or 5b and it was a typo in the text, we did not use phalloidin since it could only work in immunofluorescence (cross-linked with a phluorophore) but not in regular IHC, therefore we felt that ACE expression would be more convenient to characterize the brush border. We have corrected the mistake in the text replacing phalloidin by ACE.

6. The authors have just looked at one model with one timepoint of ischemic period and one timepoint of evaluation at 24 hours. This is inadequate to draw definitive conclusions on the role of this pathway in kidney injury. Are similar findings found with toxin models? What happens to survival of the animals over time? And what about the development of fibrosis?

The determination of the ischemic effect in renal injury and dysfunction at 24 hours is commonly used in this animal model elsewhere⁴⁻⁶. Nevertheless, and as explained in the text and figure 2, we first analyzed the onset of expression of AR in the kidney cortex after ischemic insult. As shown in the figure, we tested its expression at baseline, 1, 2, 4, 8 and 24 hours post insult finding it significantly up-regulated at 8 and 24 hours post-injury. From this, we concluded that the activation of the polyol pathway is not an early response (i.e. 15 minutes to 1 hour post insult) but rather a late response to injury but that significantly contributes to renal dysfunction as serum creatinine and BUN start to raise significantly at 8 and 24 hours but not at 4 hours (figure 3), so there is a correlation between AR up-regulation and significant renal dysfunction which can help provide a potential therapeutic window in which AR (and KHK) inhibitor may be used for the treatment of ischemic AKI (as shown in figure 9). As suggested by the reviewer and as explained above in question 1, we have expanded our knowledge of this pathway by characterizing in another model of AKI, contrast-induced nephropathy, one of the most important causes of AKI in the clinic. As shown, we demonstrate that the pathway is also activated in the proximal tubule in this model and that its blockade exerted significant protection. As indicated above, survival does not differ between wild type and fructokinase deficient mice as all animals survived the surgeries for 24 hours consistent with multiple reports elsewhere⁴⁻⁶. Furthermore, our published data demonstrate that animals survive for over 7 days post insult, of interest fibrosis is barely noticeable in this animal model at 7 and 28 days post-ischemia.

7. On what compartment is the protective effect manifest most importantly? Tubular? Vascular? Interstitial?

We thank you the reviewer for bringing up this important point. Our presented data suggest that the compartment where the protection occurs is the proximal tubule of the kidney, specifically in the S3 segment which is the more medullary segment of the PT where fructokinase is expressed. Our data cannot rule out a potential role of this pathway in endothelial cells (EC). In this regard, previous studies has been shown that upon injury AR is up-regulated in EC where it inhibits eNOS, NO release and vasodilation and that its pharmacological blockade restores oxidative stress and NO levels. Since KHK is also present in the endothelium it is likely that the activation of this pathway in afferent arterioles of the kidney can significantly contribute to the ischemic effect, the inflammation and the injury. Ideally, the deleterious role of AR, KHK and endogenous fructose production and metabolism in the endothelium versus proximal tubule could be addressed with AR and KHK floxed mice crossed with mice harboring tissue-specific (endothelium, S3 segment of the PT, etc) Cre-recombinases. Unfortunately, to the best of our knowledge, these mice have not been developed to date. Nevertheless, we have added a paragraph to the discussion section of this manuscript addressing the potential effect of the activation of this pathway in other areas besides the proximal tubule in the pathogenesis of AKI.

Reviewer #3

This manuscript hypothesizes that in mouse models of ischemic acute kidney injury (iAKI), endogenous fructose production generated by the polyol pathway is the underlying mechanism causing iAKI. The authors determined levels of standard biomarkers of renal injury in fructokinase-KO (FKKO) they previously used in related work (JASN 2014, Endogenous fructose production and fructokinase activation mediate renal injury in diabetic nephropathy), to evaluate the role of this pathway this time in iAKI, a clinically important problem with limited treatment options.

Major comments:

1. *The manuscript is well written by this group that has been very productive for years in evaluating the role of fructose in renal complications. The hypothesis as stated (endogenous ... by the polyol pathway) can be revised to better reflect the results as presented. This FKKO approach though indirect can be validated if use of aldose reductase-KO (ARKO) mice or of available AR inhibitors result in similar outcomes. In fact, an AR inhibitor ameliorates LPS inflammation-induced acute kidney injury in mice (Takahashi et al 2012 Plos One), supporting their findings.*

2. *Elimination of endogenous fructose production via AR seems to result in an outcome that may confound findings in FKKO. Ho et al 2000 Mol Cell Biol. 2000 found that AR deficiency in mice caused a partially defective urine-concentrating ability resulting in nephrogenic diabetes insipidus.*

We thank the reviewers for considering us very productive and that the manuscript is well written. The reviewer is correct in that if our hypothesis (supported by the data presented in this manuscript) is correct, AR deficient mice should be protected against AKI. Consistently, in some of our studies that relate to the activation of the polyol pathway and the endogenous production and metabolism of fructose we have employed AR deficient mice². However, AR deficient mice develop polyuria due to their inability to produce sorbitol for a proper urinary concentrating mechanism which led us to suggest that in these settings KHK would be a better target in AKI than AR blockade (we have added a new paragraph in the discussion section addressing the clinical difference between AR and KHK blockade in AKI). However, pharmacological blockade of AR –as opposed to its global deletion- could result in significant protection without causing severe nephrogenic diabetes. In this regard, besides fructokinase, luteolin has been shown to efficiently inhibit aldose reductase⁷, which we have confirmed in our proximal tubule cells. This could explain why our study demonstrate a greater protection with luteolin that the one observed in FK KO mice (figure 3 versus figure 8) suggesting that the toxicity of the polyol pathway has several components including the endogenous fructose production and its metabolism which could include the osmotic effect of sorbitol production, excessive production of NADH resulting in increased utilization of oxygen and hypoxia, and depletion of NADP by sorbitol dehydrogenase. We have expanded our discussion section with this important point brought to the reviewer acknowledging other important factors involved in the polyol pathway-dependent pathogenesis of AKI.

3. *Fig. 1 is indicated to be due to activation of the polyol pathway, but this claim can be strengthened by data on fructose consumption by patients (the excreted fructose may not be endogenous). Fig. 1 is a ratio of urinary fructose to creatinine in which both the ratio and the creatinine increase in AKI. As such, fructose concentration must be very low, especially if it is reabsorbed by the kidney. However, this manuscript did not state what method was used to determine nmol (Figs. 1, 2, 9) concentrations of fructose.*

The characteristic of the patients have been added to the methods section and AKI refer to patients with elevated serum creatinine after cardiac bypass. Unfortunately we do not possess records of dietary fructose consumption in these and control subjects so we cannot rule out differences in dietary fructose intake between control and AKI patients (a short paragraph added to the methods section in this regard). In figure 1, we calculate the urinary ratio of fructose and therefore it is normalized to *urinary creatinine but not serum creatinine*. In Aki, as a consequence of reduced glomerular filtration rate serum creatinine –and BUN- raise (thus markers of renal dysfunction) while urinary creatinine levels drop. As suggested before by reviewer 2, since we are showing a ratio, it could very well be that the net effect observed is the result of lower urinary creatinine rather than increased urinary fructose. As explained above, we have then graph urinary fructose excretion not

normalized to creatinine (i.e. urinary volume excretion). As shown in new figure 1, even without normalization, urinary fructose is significantly higher in subjects undergoing AKI compared to control subjects. We apologize for not including the methodology for urinary fructose detection. Fructose was measured biochemically with the following kit (Bioassays, EFRU250) which is based in the reaction of fructose –NADH production- by a plant enzyme not present in mammal named fructose dehydrogenase thus ensuring reduced background signal. Furthermore, and as suggested by reviewer 2 above, we have determined little or no fructokinase activity in the urine that could mask overall fructose levels in the urine. As suggested by the reviewer, new section in the methods has been added describing fructose determination in the urine.

4. Since FKKO cannot metabolize fructose, and as suggested by the manuscript fructose is continuously being produced by the polyol pathway throughout the body and renal fructose concentrations (excretion) seem very low, where is the fructose going in this mouse model that keeps on making but not eliminating fructose, and is there significant accumulation in the tissue or blood which could affect findings?

Fructose can be minimally metabolized by other sugar kinases like hexokinases (REF). These hexokinases although possess a lower affinity for fructose than fructokinase, can thus metabolize it without causing major acute ATP depletion. The observation that serum and urinary fructose is elevated in fructokinase deficient mice at baseline⁸ suggest that the role of hexokinases in metabolizing endogenous fructose is minimal though. It is important to note, that aldose reductase is a rate-limiting enzyme whose baseline expression is relatively low in other tissues including liver and kidney cortex (figure 2) and therefore the endogenous production of fructose is minimal –although present as it can be observed in fructokinase deficient mice at baseline- and therefore we do not anticipate significant affection of our findings by baseline fructose. However, upon stimulation (diabetes, hypoxia, oxidative stress, etc), AR expression is up-regulated and fructose endogenously produced and metabolized.

5. The finding that the flavonoid luteolin is also a FK inhibitor is exciting and innovative, but this compound has numerous biological effects (reviewed by Lopez-Lazaro 2009 Mini Rev Med Chem). The authors' interesting finding of FK inhibition requires validation of luteolin specificity. Because they are the leading investigators in fructose physiology, their work will likely be cited by other workers and the public may consume commercially available luteolin while partaking of fructose laden foods. The authors should discuss potential side effects of luteolin in other organ systems, and off target effects in the kidney, especially on other related kinases. Since the drug is given intravenously, these indirect effects may confound findings presented in Figs. 8 & 9. 6. It is not clear how 941 nM can statistically be the IC50, as there seems to be only one point between 0 and 100% inhibition (Fig. 8A). What is the standard error for this estimate? What is the rationale for the 2.5 mg/kg in vivo dose?

We agree with the reviewer in the variety of off-target effects of luteolin. In this regard, AR has been found to inhibit efficiently AR as well thus exacerbating its potential protective response in AKI (figure 8). As suggested by the reviewer, we have added a paragraph in the discussion section detailing potential off target effects of luteolin on other enzymes and kinases and how that can affect our presented data. Importantly, we have recently published a manuscript in which we identified plant-derived active compounds with efficient fructokinase inhibitory activity⁹. In this manuscript we provide the specific details to determine khk activity. Of these, osthole, a derivative obtained from plants of the Angelica family, demonstrated one of the greatest inhibitory activity and recently, Zheng et al, demonstrated its efficiency in ameliorating renal injury in mice undergoing ischemic AKI¹⁰. While the specific mechanism whereby osthole exerted protection in AKI was not

proposed, we expect based on our data that it could be at least partially mediated by fructokinase inhibition in the renal cortex.

Minor concern:

1. Since this is a general and not a nephrology journal, readers should be guided accordingly as to where the tubular dilatation and cast formation are in Fig. 4, at least with arrows and directions. Similar layperson explanations should be made in Figs. 5, 6, 8 and 9. Fig. 2 was well presented.

We agree with the reviewer and therefore, we have updated the figures pointing the areas of injury in figures 4, 8 and new figure 10.

- 1 Nakayama, T. *et al.* Dietary fructose causes tubulointerstitial injury in the normal rat kidney. *Am J Physiol Renal Physiol* **298**, F712-720, doi:10.1152/ajprenal.00433.2009 (2010).
- 2 Lanaspá, M. A. *et al.* Endogenous fructose production and metabolism in the liver contributes to the development of metabolic syndrome. *Nat Commun* **4**, 2434, doi:10.1038/ncomms3434 (2013).
- 3 Roncal-Jimenez, C. *et al.* Heat Stress Nephropathy From Exercise-Induced Uric Acid Crystalluria: A Perspective on Mesoamerican Nephropathy. *Am J Kidney Dis* **67**, 20-30, doi:10.1053/j.ajkd.2015.08.021 (2016).
- 4 Ahuja, N. *et al.* Circulating IL-6 mediates lung injury via CXCL1 production after acute kidney injury in mice. *Am J Physiol Renal Physiol* **303**, F864-872, doi:10.1152/ajprenal.00025.2012 (2012).
- 5 Andres-Hernando, A. *et al.* Splenectomy exacerbates lung injury after ischemic acute kidney injury in mice. *Am J Physiol Renal Physiol* **301**, F907-916, doi:10.1152/ajprenal.00107.2011 (2011).
- 6 Klein, C. L. *et al.* Interleukin-6 mediates lung injury following ischemic acute kidney injury or bilateral nephrectomy. *Kidney Int* **74**, 901-909, doi:10.1038/ki.2008.314 (2008).
- 7 Wang, Q. Q., Cheng, N., Zheng, X. W., Peng, S. M. & Zou, X. Q. Synthesis of organic nitrates of luteolin as a novel class of potent aldose reductase inhibitors. *Bioorg Med Chem* **21**, 4301-4310, doi:10.1016/j.bmc.2013.04.066 (2013).
- 8 Ishimoto, T. *et al.* Opposing effects of fructokinase C and A isoforms on fructose-induced metabolic syndrome in mice. *Proc Natl Acad Sci U S A* **109**, 4320-4325, doi:10.1073/pnas.1119908109 (2012).
- 9 Le, M. T. *et al.* Bioactivity-Guided Identification of Botanical Inhibitors of Ketohexokinase. *PLoS One* **11**, e0157458, doi:10.1371/journal.pone.0157458 (2016).
- 10 Zheng, Y. *et al.* Osthole ameliorates renal ischemia-reperfusion injury in rats. *J Surg Res* **183**, 347-354, doi:10.1016/j.jss.2013.01.008 (2013).

PEER REVIEW FILE

Reviewers' comments:

Reviewer #2 (Remarks to the Author):

This is a revised version of the originally submitted manuscript on the protective role of fructokinase blockade in the pathogenesis of ischemic AKI in mice. The authors have adequately responded to many of the concerns. They have added a new model of AKI- a contrast induced nephropathy model.

The authors have found, using inhibitors and genetic approaches that fructokinase inhibition is protective.

An important limitation of the study is the fact that only time points during the first 24 hours post ischemia are studied. Most of the important things that happen with ischemia or contrast relate to longer-term effects and not necessarily events that occur in the first 24 hours. The authors do not seem to be concerned about this but I do think it important enough for the authors to recognize it as a limitation in the manuscript. The authors comment that this model does not lead in their hands to fibrosis. This may be due to the fact that they are only doing 22 min of ischemia. If it does not lead to fibrosis, then how does the model related to their comments regarding Mesoamerican nephropathy which they believe to relate to repeated bouts of AKI leading to CKD?

The authors should be explicit about whether or not their control mice received vehicle for the contrast agents in their model of contrast induced nephropathy

Reviewer #3 (Remarks to the Author):

The authors have for the most part addressed this reviewer's comments, except in two areas where both the ms and the authors' responses were unclear.

1. They have eliminated an apparently statistically incomplete graph (original Fig. 8A) but retained its IC50 results in the text with claims related to potency.
2. The authors are also unclear about blood and tissue concentrations of FKKO, as their response (Fig. 2) has no direct measurements of these readouts that should be available from studies on FKKO humans and mice showing fructose concentrations that hexokinase may become relevant at some point.

Response to reviewers:

Reviewer #2

An important limitation of the study is the fact that only time points during the first 24 hours post ischemia are studied. Most of the important things that happen with ischemia or contrast relate to longer-term effects and not necessarily events that occur in the first 24 hours. The authors do not seem to be concerned about this but I do think it important enough for the authors to recognize it as a limitation in the manuscript. The authors comment that this model does not lead in their hands to fibrosis. This may be due to the fact that they are only doing 22 min of ischemia. If it does not lead to fibrosis, then how does the model related to their comments regarding Mesoamerican nephropathy which they believe to relate to repeated bouts of AKI leading to CKD?

In this revised version we have now added to the discussion section a paragraph acknowledging the limitation of a short study. By working on a 24-hour model, the reviewer is correct in that we are missing important pathogenic mechanisms leading to the development and progression of chronic kidney disease (CKD) that does not occur within the first 24 hours as it would occur in Mesoamerican nephropathy or aging-associated kidney disease. The idea that CKD could be result of multiple uncontrolled episodes of AKI is really novel and a hot topic research in nephrology. In this regard, we want to emphasize that the goal of our study is to address the importance of endogenous fructose production and its metabolism in models of acute (as opposed to chronic) kidney injury. How AKI and whether the activation of the metabolism of fructose is relevant for the chronicity of the disease will be the focus of further studies. However, it is important to note that we already have presented evidence of the importance of this pathway in other models of chronic kidney disease including diabetic nephropathy¹, heat/dehydration² and more recently, aging-associated renal disease³. In this manuscript, we provide evidence of how early this events may occur and its relative importance for long term effects and the chronicity of AKI into CKD.

The authors should be explicit about whether or not their control mice received vehicle for the contrast agents in their model of contrast induced nephropathy

We thank the reviewer for bringing this up. Control mice on CIN received proper controls and volumes for each drug that was administered in this model (L-NAME, Indomethacin and contrast). We have edited the results and methods section to include the use of proper vehicles in control mice in the model of CIN.

Reviewer #3

They have eliminated an apparently statistically incomplete graph (original Fig. 8A) but retained its IC50 results in the text with claims related to potency.

We apologize for the mistake. As indicated in our previous response, since the graph was incomplete we decided to remove it and indicated the obtained IC50 from our in vitro assay as we described in here⁴. The new IC50 (11.9 μ M) is now shown in the text.

The authors are also unclear about blood and tissue concentrations of FKKO, as their response (Fig. 2) has no direct measurements of these readouts that should be available from studies on FKKO humans and mice showing fructose concentrations that hexokinase may become relevant at some point.

Figure 9 show urinary fructose excretion in wild type and fructokinase knockout animals undergoing iAKI. We agree with the reviewer that blood and tissue levels of sorbitol and fructose should be included as well. In this regard we have already shown that fructokinase knockout mice tend to have greater serum fructose levels than wild type animals even in the absence of fructose intake as a result of lack of metabolism of endogenously produced fructose⁵. Our data indicates that in both wild type and fructokinase deficient mice there is an activation of the polyol pathway and endogenous fructose production upon ischemic insult. Although the difference was found to be not significant between wild type and fructokinase knockout animals, there is a slight tendency of greater levels in fructokinase knockout mice. As suggested by the reviewer, the data including the mean and standard deviation is now included in the results section.

- 1 Lanaspá, M. A. *et al.* Endogenous fructose production and fructokinase activation mediate renal injury in diabetic nephropathy. *J Am Soc Nephrol* **25**, 2526-2538, doi:ASN.2013080901 [pii]
10.1681/ASN.2013080901 (2014).
- 2 Roncal Jimenez, C. A. *et al.* Fructokinase activity mediates dehydration-induced renal injury. *Kidney Int*, doi:ki2013492 [pii]
10.1038/ki.2013.492 (2013).
- 3 Roncal-Jimenez, C. A. *et al.* Aging-associated renal disease in mice is fructokinase dependent. *Am J Physiol Renal Physiol* **311**, F722-F730, doi:10.1152/ajprenal.00306.2016 (2016).
- 4 Le, M. T. *et al.* Bioactivity-Guided Identification of Botanical Inhibitors of Ketohexokinase. *PLoS One* **11**, e0157458, doi:10.1371/journal.pone.0157458 (2016).
- 5 Ishimoto, T. *et al.* Opposing effects of fructokinase C and A isoforms on fructose-induced metabolic syndrome in mice. *Proc Natl Acad Sci U S A* **109**, 4320-4325, doi:1119908109 [pii]
10.1073/pnas.1119908109 (2012).